Expected mortality risk for coho salmon landed in recreational troll fisheries using 1/0 and 6/0 hooks in the marine waters of Washington state

Scordino Jonathan J. jonathan.scordino@makah.com 1
Walsh Ryan P. 1
Jasper William 1
Roche Deon J. 1
Tyler William 1 2
1 Makah Fisheries Management, Makah Tribe , Neah Bay , WA , United States of America
2 Clallam Bay High School , Clallam Bay , WA , United States of America
Yapıcı Sercan
Electronic publication date: 2025 May 15
Publication date: 2025
Volume: 13
Electronic Location ID: e19434
Received 2024 Nov 4; Accepted 2025 Apr 16
Copyright: ©2025 Scordino et al.
Copyright year: 2025
Copyright holder: Scordino et al.
License: This is an open access article distributed under the terms of the Creative Commons Attribution License, which permits unrestricted use, distribution, reproduction and adaptation in any medium and for any purpose provided that it is properly attributed. For attribution, the original author(s), title, publication source (PeerJ) and either DOI or URL of the article must be cited.
License URL: https://creativecommons.org/licenses/by/4.0/

Keywords: Coho salmon, Oncorhynchus kisutch, Recreational fishing, Hooking mortality, Fisheries management, Salmon, Chinook salmon, Pink salmon

Funding: The SeaDoc Society, a program of the Karen C. Drayer Wildlife Health Center, School of Veterinary Medicine, University of California, Davis SeaDoc Society Tribal and First Nations Grants Project number A23-3415-S001 This work was supported by the SeaDoc Society, a program of the Karen C. Drayer Wildlife Health Center, School of Veterinary Medicine, University of California, Davis. The funding award was through the SeaDoc Society Tribal and First Nations Grants (Project number A23-3415-S001). The funders had no role in study design, data collection and analysis, decision to publish, or preparation of the manuscript.

==============================
Recreational salmon fisheries in the state of Washington are managed with size-selective and mark-selective rules to promote the release of wild and undersized salmon. In order for this management approach to be effective, the fishery must have low mortality of hooked and released salmon. In this study, we directly compared the fishing performance of small (1/0, 14.36 mm gap) and large (6/0, 17.20 mm gap) hooks in recreational salmon troll fisheries with the goal of evaluating differences in drop-off rates, size of salmon caught, and the probability a fish would be hooked in a region documented to have low, intermediary, or high risk of mortality. We found strong evidence that hook size affects the probability that an angler can land a fish with drop-off rates of 48.9% for fish hooked on small (1/0) hooks and 34.0% for fish hooked on large (6/0) hooks. There were no significant differences in the average length of pink, coho, or Chinook salmon caught on small (1/0) or large (6/0) hooks. Likewise, we found no significant differences in the proportion of coho and Chinook catch that were longer or shorter than their respective legal-size limits. Hook size did significantly affect the expected mortality risk of coho salmon with large (6/0) hooks being 1.82 times more likely to hook a salmon in body regions associated with high or intermediary risk of mortality than small (1/0) hooks. No significant differences were observed in the hooking rates of Chinook and pink salmon in body regions with high, intermediary, or low expected mortality risk, however our confidence in this finding is limited due to sample size. It is important to carefully assess whether these results support restrictions on hook sizes used in recreational coho salmon troll fisheries. The observation of reduced expected mortality risk of coho salmon landed using small (1/0) hooks as compared to when using large (6/0) hooks highlights the importance of evaluating the effect of hook size during recreational salmon troll fisheries on post-release survival and will hopefully stimulate more research on the effects of hook size and modeling on whether hook size regulations can improve salmon conservation.

Introduction

Salmon management in Washington state is challenged by balancing the goals of providing fishing opportunities for the culture (Amberson et al., 2016; Efford et al., 2023), economics (Gislason et al., 2017), and recreation of the region (Beaudreau & Whitney, 2016) while achieving the goals of promoting healthy salmon runs and the recovery of depleted salmon populations. Washington state utilizes hatcheries to help offset the loss habitat for salmon, aid in recovery of wild stocks, and to enhance fishing opportunities. Coho (Oncorhynchus kisutch) and Chinook (Oncorhynchus tshawytscha) salmon reared at hatcheries in Washington state generally have their adipose fin cut off prior to release, allowing anglers and fishery managers to differentiate hatchery-origin and wild-origin salmon. The marking of coho and Chinook salmon allows managers at Washington Department of Fish and Wildlife to implement ‘mark-selective’ fishery regulations that promote the retention of hatchery-origin salmon while wild-origin salmon are released. In addition to regulations requiring that recreational fisheries target hatchery-origin salmon, the state of Washington also has daily catch limits and minimum size limits for coho and Chinook salmon (Washington State Department of Fish and Wildlife, 2022) to increase the availability of large coho and Chinook for anglers (Waters & Huntsman, 1986; Savitz et al., 1995).

Despite regulations requiring the release of wild and undersized coho and Chinook salmon, recreational fisheries can still have a large impact on wild salmon runs of concern because not all of the released salmon survive capture and release. A meta-analysis of hooking studies found that on average 27.4% (SE = 6.6) of coho salmon and 21.2% (SE = 3.7) of Chinook salmon die post-release due to injuries sustained by hooking and handling (Hühn & Arlinghaus, 2011). In addition to the observed mortality rate of landed salmon, an unknown proportion of hooked salmon that escape the hook prior to landing (drop-offs) also die due to the hooking injuries they sustain (Lawson & Sampson, 1996). Lastly, it is possible for a wild salmon to be caught and released on multiple occasions in mark-selective fisheries as it travels from its ocean feeding grounds to its natal river. The conservation benefits of mark-selective and size-selective fishing regulations would be improved if hooking mortality were minimized.

One possible measure to increase survival of hooked-and-released salmon in recreational salmon fisheries is to regulate the size of hooks used (Muoneke & Childress, 1994; Patterson et al., 2017; Courter et al., 2023). Most research on mortality of hooked-and-released salmon have not directly evaluated the effect of hook size used on the survival of hooked and released salmon. Studies using small (1/0–2/0) hooks have found lower mortality rates (Muoneke & Childress, 1994) for hooked Chinook and coho salmon than were found for Chinook and coho salmon caught during studies using large (5/0–7/0) hooks (Butler & Loeffel, 1972; Wertheimer, 1988; Wertheimer et al., 1989; Orsi, Wertheimer & Jaenicke, 1993). However, one study with an extremely high estimated post-release mortality (69%) used 2/0 hooks showing that the use of smaller hooks may not reliably reduce post-release mortality rates (Vincent-Lang, Alexandersdottir & McBride, 1993). The one study that did directly compare mortality rates between two hook sizes used in salmon fishing compared two hooks of rather similar size (size 2 and size 4 treble hooks) and found no effect of hook size on mortality rate (Savitz et al., 1995). Together these past studies do not paint a clear picture on whether hook size affects survival of caught and released salmon. However, they support additional studies that directly assess hook size and survival in salmon fisheries after introducing the potential that hook size can influence survival. In addition to improving the survival of hooked and released salmon, regulating hook size may also provide other management benefits such as improving the size-selectivity of the fishery (Muoneke & Childress, 1994), reducing catch rates of non-target species (Petersen et al., 2020), and reducing drop-off rates. Past research has found that the handling of salmon post capture, environmental variables like water temperature, and the body location a fish is hooked can have strong influences on post-release survival and the stress experienced by fish that live through the capture (Muoneke & Childress, 1994; Hühn & Arlinghaus, 2011; Twardek et al., 2018; Cooke et al., 2019; Fritts et al., 2023; Lunzmann-Cooke et al., 2024). This work will build on past research by focusing on the very important parameter of capture injury due to hook size.

This study investigates the fishing performance of small (1/0, 14.36 mm gap) and large (6/0 16.40 mm gap) hooks in recreational trolling fisheries, focusing on three key parameters. First, we compared the effect of hook size on the frequency of hooking anatomical regions documented to have low, intermediary, and high risk of expected post-release mortality (Wertheimer et al., 1989) using the evaluation approach used by Orsi, Wertheimer & Jaenicke (1993). The relative risk of mortality from hooking related injuries for anatomical regions documented by Lunzmann-Cooke et al. (2024) supports the rates published by Wertheimer et al. (1989), with hooking of the eye causing more frequent post-release mortality than areas of the upper and lower jaws. Past research suggests that hook size influences the severity of injury and risk of morality of hooked fish (Muoneke & Childress, 1994), primarily via the effect of hook size on hooking location (Arlinghaus et al., 2007; Patterson et al., 2017). Accordingly, we hypothesized that small (1/0) hooks would penetrate salmon in body regions with intermediary or high post-release mortality risk less frequently than large (6/0) hooks due to the greater distance (gap) between the hook tip and shank in the larger hook, allowing it to penetrate deeper in the mouth and damage organs such as the gills and eye. Second, we evaluated if using a small hook would increase drop-off rates as compared to large hooks. Based on discussions with anglers who utilize larger hooks to improve catch retention, we hypothesized that the large (6/0) hook would have a significantly lower drop-off rate than the small (1/0) hook. Third, we evaluated if hook size affects the size of salmon caught. Based on past studies of salmon fishing (Orsi, 1987; Orsi, Wertheimer & Jaenicke, 1993), we hypothesized that larger hooks would catch larger coho and Chinook salmon and reduce the hooking rates of undersized (sub-legal) coho and Chinook salmon. This study focuses on troll fishing because it is the most commonly employed recreational salmon fishing technique in the marine environments of the state of Washington and Salish Sea region (Diewert, Nagtegaal & Patterson, 2002), especially for those fishing from private boats. The overall objective of this study is to evaluate how hook size influences the post-release mortality risk, drop-off rates, and the size of salmon caught in recreational salmon troll fisheries. If hook size is found to reduce mortality and drop-off rates, it could lead to new management recommendations regarding hook regulations in recreational salmon troll fisheries.

Materials & Methods

Gear assembly

The objective of this study was to compare the fishing performance of large and small hooks in recreational troll fisheries in Washington state marine waters. Terminal fishing gear was chosen for this project and assembled with the goal of controlling all variables except for the hook sizes being compared. We chose to use a 1/0 hook (14.36 mm gap) for our small hook and a 6/0 hook (16.40 mm gap) as our large hook. Our rationale was that the 1/0 and 6/0 hooks bookend the typical range of hook sizes used by recreational anglers in Washington state and that comparing the largest and smallest of the typical size range used by anglers would give us the best chance of documenting hook size effects if they exist. We used the same brand of octopus hook (Gamakatsu) for both 1/0 and 6/0 hooks (Fig. 1). This approach ensured that any observed differences in fishing performance were due solely to hook size, and not to variations in design between different manufacturers. The 6/0 hooks (stock number 02416-25) were de-barbed with pliers while the 1/0 hooks (stock number 75411-25) were already barbless. The measurements of the two hooks are presented in Fig. 1.

Figure 1 We compared the fishing performance of Gamakatsu octopus 1/0 (stock number 02416-25) and 6/0 (stock number 75411-25) hooks during recreational salmon troll fishing.

Important measurements of each hook are listed in the figure.

We chose to fish exclusively with hoochies rather than using bait or other common salmon lures such as plugs or spoons (Orsi, Wertheimer & Jaenicke, 1993) because using the hoochie provided us the ability to control the position of the hook relative to the lure. For example, if a salmon were to be hooked on a spoon, the hooking region might differ from a 1/0 to 6/0 hook due to the dimensions of the hook and the point of the hook being closer to the spoon with a small hook than with a large hook. We rigged each hoochie with a single hook positioned with the bend of the hook at the tail end of the skirt of the hoochie by placing beads and in some cases skirts as spacers between our hook eye and the head of the hoochie; the small (1/0) hooks required the use of more beads than did the large (6/0) hooks. Another benefit of using hoochies is that the size of the hook had minimal or no effect on the swimming action of the lure (Fig. 2).

Figure 2 Hoochies were used as the artificial lure for comparing the fishing performance of 1/0 and 6/0 hooks in recreational salmon trolling in the marine waters of northwest Washington.

Each hoochie was configured in the same way with a 106.7 cm leader of 27 kg monofilament line and with the bend of each hook even with the tail end of the hoochie skirt. The position of the hook in the hoochie was controlled by placing beads and sometimes skirts as spacers within the hoochie. A US quarter is included for size perspective (diameter = 2.4 cm).

Each hoochie was attached to a 106.7 cm leader that was attached to a 27.9 cm flasher. We used Maxima Fishing Line’s Ultragreen 27 kg test monofilament line for the leader. The purpose of using 27 kg test monofilament leader was to allow the movement of the flasher to give a swimming action to the hoochie while being trolled. Gear set ups (flasher and hoochie) for both small (1/0) and large (6/0) hooks were observed in shallow water while the boat traveled at trolling speeds (∼2–4 knots) and no differences were observed in the swimming action of lures for gear set ups with a large or a small hook.

We fished with two different sizes of hoochies (5.7 cm and 10.8 cm) and in twelve different colors resulting in 16 different color and size combinations of hoochies. Having 16 color and size combinations allowed us to optimize our fishing success by fishing the color and size of lure that we expected fish to prefer while we fished.

Field methods

All fishing effort occurred in the marine waters of northwest Washington in Marine Areas 4B and 5 in the Strait of Juan de Fuca and in Marine Area 4 of the Pacific Ocean during June through September of 2023 and 2024 (Fig. 3). We fished aboard two vessels during the experiment. The Optimistic is a 5.2 m Arima Sea Chaser recreational fishing vessel. The R/V 1 is an 8.5 m vessel made by Pacific Skiffs that serves as the Makah Tribe’s research vessel. Both vessels were equipped with electric downriggers near the stern along their port and starboard rails. All electric downriggers were attached to a 6.8 kg lead downrigger ball with a downrigger cable. Release clips were attached to the downrigger cable via a cable snap. These release clips were clipped to the line on fishing poles with the flasher and hoochie about five m from the clip. The use of the downrigger allowed us to troll our fishing gear at a specific and accurate depth while the boat traveled at 2–4 knots. We typically fished one pole on each side of the boat but sometimes stacked release clips roughly six m apart on the same downrigger cable to allow fishing of four poles simultaneously (two poles on each downrigger). All fishing activity was conducted following Washington state sport fishing regulations by the authors and volunteer anglers. A total of 34 anglers (four authors and 30 volunteers) fished during the study, ranging in experience from first-time fisher to expert angler.

Figure 3 Map of the study area where research was conducted during June through September of 2023 and 2024.

The map includes Marine Areas for management by Washington Department of Fish and Wildlife and dots representing where troll fishing effort started for each set of the experiment to compare the fishing performance of 1/0 and 6/0 hooks in recreational salmon troll fisheries.

We recorded the set time, fishing depth, hook size, and the color and size of the hoochie used during each set of our fishing gear. We recorded the end time and whether or not a fish was hooked whenever we retrieved our fishing gear. We were able to tell when a fish was hooked by watching for a change in motion of the tip of the pole attached to the downrigger release clip. Larger fish were typically able to trip the release clip, freeing the fishing line from the downrigger cable; for smaller fish, the angler pulled up swiftly on the pole to disengage the release clip.

When a fish was hooked, one crew member reeled the fish in while other crew members netted the fish with a landing net and recorded data. We recorded whether or not the fish was landed (brought aboard the vessel). If the fish was successfully landed, then we quickly examined the fish to determine its species. Air exposure and handling times were limited for each landed fish to minimize stress and reduce risk of post-release mortality (Nguyen et al., 2013; Lunzmann-Cooke et al., 2024). If the fish was a species of salmon, then we determined the anatomical region where the salmon was hooked and whether or not the salmon was bleeding. We utilized body regions defined by Wertheimer (1988); Fig. 4). If a salmon had evidence of the hook penetrating multiple anatomical regions, the region with the greatest observed mortality rate according to Wertheimer et al. (1989); Table 1) was recorded. For instance, if a salmon was hooked through the maxilla and eye, then we recorded that the salmon was hooked in the eye. We recorded that a salmon was bleeding if it had a steady flow of blood and did not record it as bleeding if it had no signs of blood or if it had just a spotting of blood (our definition of bleeding aligns with bleeding categories 1 and 2 of Lunzmann-Cooke et al. (2024) of notable (category 1) to significant (category 2) bleeding). A crew member then removed the hook and transferred the salmon to a measuring trough filled with water where we measured its fork length (FL) from the tip of the snout to the notch of the tail. We recorded the time we reeled up and that no fish was hooked when we landed a non-salmon fish species because our research goal was to assess catch performance for our target species of Chinook, coho, and pink (Oncorhynchus gorbuscha) salmon.

Figure 4 Field guide for determining which body region was hooked during this study.

Body regions assessed were defined by Wertheimer et al. (1989) and have published estimates of mortality rates of Chinook salmon hooked in the region. If the salmon was hooked in a body region not otherwise identified in this figure, then it was classified as hooked on the body.

Table 1 Mortality risk scores for salmon hooked by body region.

The risk scores were based on observed mortality rates for large (>66 cm FL) and small (<66 cm FL) Chinook salmon for the body region hooked with a barbed 6/0 hook in a commercial troll fishery (Wertheimer et al., 1989). Mortality risk was binned into classifications of low, intermediary, and high risk of mortality corresponding to mortality rates observed by Wertheimer et al. (1989) of 0–15%, 15–50%, and greater than 50%, respectively.

Body region hooked	Mortality rate for large Chinook	Risk of mortality	Mortality rate for small Chinook	Risk of mortality	
Snout	9.7%	Low	0.0%	Low	
Maxillary	10.9%	Low	4.0%	Low	
Lower Jaw/Tongue	8.3%	Low	17.6%	Intermediary	
Corner of mouth	7.7%	Low	5.4%	Low	
Eye	21.2%	Intermediary	21.9%	Intermediary	
Isthmus	38.2%	Intermediary	29.2%	Intermediary	
Gill	85.0%	High	85.0%	High	
Cheek	15.7%	Intermediary	15.4%	Intermediary	

Data analysis

Summary statistics were computed for days fished, hours fished for each hook size, and numbers of salmon caught by species.

We tested our hypothesis that large (6/0) hooks would catch larger salmon than small (1/0) hooks using two approaches. First, comparisons were made for the average length of salmon caught on the two hook sizes for each species using T-tests. Second, for coho and Chinook salmon, we also used chi-squared tests to compare categorical variables, allowing us to evaluate if the proportion of legal and sublegal sized fish differed by hook size. The size limits used were 40.6 cm for coho and 55.9 cm for Chinook based on Washington state regulations in Marine Area 5 in 2023 and 2024.

Drop-off rates for the two hook sizes were compared using a chi-squared test to evaluate our hypothesis that larger hooks will have lower drop-off rates.

Last, we tested our hypothesis that salmon hooked on small (1/0) hooks would have lower expected mortality risk than those caught on large (6/0) hooks through two approaches. In this study we did not directly observe if a hooked salmon lived or died due to its hooking injury through post-capture monitoring. Instead, we utilized the approach of Orsi, Wertheimer & Jaenicke (1993) of using published mortality rates from Wertheimer et al. (1989) to inform the expected risk of post-release mortality of salmon hooked in our study. Wertheimer et al. (1989) found that mortality rates for each hooked anatomical region were different for small (<66 cm FL) and large (>66 cm FL) Chinook salmon using barbed 6/0 hooks. For this study, we assumed that all species of salmon caught on barbless 1/0 and 6/0 hooks during recreational troll fishing would have similar risk of mortality by body region hooked as were observed by Wertheimer et al. (1989) for Chinook salmon caught during commercial salmon trolling operations with barbed 6/0 hooks. Furthermore, we assumed for this analysis that hook location was the only determinant for mortality risk and ignored other factors to affect survival such as air exposure, water temperature, capture depth, angler experience, and scale loss (Muoneke & Childress, 1994; Hühn & Arlinghaus, 2011; Nguyen et al., 2013; Cook et al., 2018; Lunzmann-Cooke et al., 2024). We assigned each body region a qualitative risk of expected mortality as either high, intermediary, and low based on the results of Wertheimer et al. (1989) rather than using their observed mortality rates for our analysis. We defined a body region as having high expected mortality risk if Wertheimer et al. (1989) observed a mortality rate of greater than 50%, body regions were defined as intermediary expected mortality risk for regions that Wertheimer et al. (1989) observed to have a 15–50% mortality rate, and body regions were defined as having a low expected mortality rate for regions (Wertheimer et al., 1989) observed to have a mortality rate of less than 15%. Body regions of low, intermediary, and high expected mortality risk for large (>66 cm FL) and small (<66 cm FL) salmon are listed in Table 1 with observed mortality rates from Wertheimer et al. (1989). Fish hooked outside of the head region were not included in the analysis because we did not have estimated mortality rates available. Other studies that have documented mortality risk for salmon by hooking region generally confirm the qualitative scores we have used (Orsi, Wertheimer & Jaenicke, 1993; Lunzmann-Cooke et al., 2024) with the notable exception of the eye region which was found to have no effect on mortality during in-river recreational salmon fisheries (Lindsay et al., 2004; Fritts et al., 2023). Catch records for Chinook, coho, and pink salmon species were summarized in 3 × 2 contingency tables for low, intermediary, and high expected mortality risk by hook sizes of 1/0 and 6/0. Fisher’s exact test was used to compare contingency tables to test our hypothesis that small (1/0) hooks would be less likely to hook salmon in body regions reported to have high or intermediary risk of mortality as compared to large (6/0) hooks. Fisher’s exact tests were used where expected categorical frequencies were less than five. An odds ratio test comparing hooking rates of body regions with low risk of expected mortality to body regions with high and intermediary mortality risk pooled together was used to evaluate the effect size for species with significantly different proportions of expected mortality risk scores between the two hook sizes. Last, we used chi-squared tests to compare the proportion of each salmon species caught that were bleeding for 1/0 and 6/0 hooks.

Results

We conducted 55 days of fishing effort in Marine Areas 4, 4B, and 5 in the waters of northwest Washington (Fig. 3). The number of poles fished each day ranged from 1 to 4; it was most common for us to fish two poles. In total we had 128 h and 24 min of fishing effort with 1/0 hooks and 149 h and 34 min with 6/0 hooks. We hooked 282 fish on the 6/0 hook (1.9 hook-ups per hour) and 317 fish on the 1/0 hook (2.5 hook-ups per hour). Of the total fish hooked, 187 salmon were landed to the boat on 6/0 hooks (1.24 salmon landed per hour) and 162 salmon on 1/0 hooks (1.26 salmon landed per hour). Of the salmon caught on the 6/0 hook, 35 were Chinook, 134 were coho, and 18 were pink. Of the salmon caught on the 1/0 hook, 35 were Chinook, 113 were coho, and 14 were pink salmon (Table 2). We found strong evidence that hook size affected drop-off rates (X2 = 13.53, df = 1, p < 0.001) with observed drop-off rates of 48.9% on 1/0 hooks and 34.0% on 6/0 hooks. Drop-off rates were 1.85 times greater for fish hooked on a 1/0 hook than when hooked on a 6/0 hook (odds ratio test 95% confidence interval 1.33–2.58).

Chinook, coho, and pink salmon caught on 1/0 hooks were all on average larger than those caught on 6/0 hooks, but the differences were not statistically significant (Table 2). We found no evidence for differences in the proportions of the catch of coho and Chinook salmon that were of legal size, > 40.6 cm and > 55.9 cm respectively, between 1/0 and 6/0 hooks (Table 2).

We observed significant differences in the proportion of coho salmon hooked in anatomical body regions of low, intermediary, and high risk between coho salmon hooked on large (6/0) and small (1/0) hooks (Fisher’s exact test, p = 0.029). Coho salmon caught on a large (6/0) hook were 1.82 times more likely to be hooked in a body region associated with high or intermediary risk of expected mortality than were coho hooked with a small (1/0) hook (odds ratio test 95% confidence interval 1.09–3.05). Chinook and pink salmon, on the other hand, did not have evidence of statistical differences in expected mortality risk scores between the two evaluated hook sizes (Table 3). We found no evidence for differences in probability that a salmon hooked on a small (1/0) hook was more likely to be bleeding than a salmon hooked on a large (6/0) hook for either Chinook or coho salmon. We did, however, find strong evidence that pink salmon are more likely to be observed bleeding if caught with a small (1/0) hook than with a large (6/0) hook (Table 2; Fisher’s exact test, p = 0.002).

Table 2 Summary statistics for salmon caught in the study.

The average length is the fork length in cm with ± the standard error. Chinook salmon greater than 55.9 cm and coho salmon greater than 40.6 cm were classified as legal sized; there is no size limit for retention of pink salmon in Washington state. Mortality risk due to hooking injury was classified as “low” for mortality risk of 0–15%, “intermediary” for mortality risk of 15–50%, and “high” for mortality risk greater than 50% based on published rates for the body region where the salmon was hooked (Wertheimer et al., 1989). All salmon were caught on 1/0 (14.36 mm gap) and 6/0 (16.40 mm gap) octopus hooks during recreational salmon trolling in the marine waters of northwest Washington in 2023 and 2024.

Species	Coho salmon		Chinook salmon		Pink salmon	
 	1/0	6/0		1/0	6/0		1/0	6/0	
Total Caught	113	134		35	35		14	18	
% Legal size	95.6%	92.5%		51.4%	40.0%		–	–	
% Bleeding	21.5%	27.5%		20.6%	17.6%		42.9%	0.0%	
% Low risk	62.8%	48.1%		54.3%	48.6%		28.5%	33.3%	
% Intermediary risk	36.3%	47.4%		42.9%	48.6%		71.5%	66.7%	
% High risk	0.9%	4.5%		2.9%	2.9%		0.0%	0.0%	
Average length (cm)	52.6 ± 0.6	51.7 ± 0.7		55.0 ± 2.1	52.5 ± 1.9		49.9 ± 1.2	48.8 ± 1.1	

Table 3 Count of large (> 66 cm FL) and small (<66 cm FL) salmon caught by species hooked with either 1/0 or 6/0 hooks by anatomical region the hook penetrated.

If a hook point was observed to have passed through two regions, then the region with the highest risk of mortality was recorded. Salmon recorded with an unknown body region had come off the hook and hooking location could not be determined. All salmon were caught during recreational salmon troll fishing in northwest Washington during 2023 and 2024.

	Chinook		Coho		Pink	
	Large	Small		Large	Small		Small	
Anatomical Region	1/0	6/0	1/0	6/0		1/0	6/0	1/0	6/0		1/0	6/0	
Snout	0	2	3	5		0	1	9	11		1	0	
Maxilla	3	0	3	4		2	0	17	21		1	1	
Lower jaw	1	2	10	6		0	0	21	24		5	8	
Corner of mouth	2	1	7	3		1	0	40	28		2	5	
Eye	0	0	2	8		0	1	11	28		3	2	
Tongue	0	0	3	2		0	0	2	2		1	1	
Isthmus	0	0	0	0		0	0	4	6		0	0	
Gills	0	0	0	0		0	0	0	5		0	0	
Body	0	0	0	0		0	0	2	3		0	0	
Cheek	0	1	0	0		0	0	3	2		1	1	
Unknown	0	0	1	1		0	0	1	2		0	0	
Total	6	6	29	29		3	2	110	132		14	18	

Discussion

Salmon fisheries in the state of Washington are managed to be mark-selective and size-selective with anglers being required in most management areas to release sublegal length Chinook and coho salmon and all Chinook or coho salmon with an intact adipose fin. For size-selective and mark-selective fishery regulations to be effective management strategies, the fishery has to have low mortality rates for hooked and released fish (Savitz et al., 1995). The objective of this study was to determine if regulations on the hook sizes used in recreational troll fisheries in the marine waters of Washington state could improve survival of hooked and released salmon. We found that coho salmon were significantly more likely to be hooked in anatomical regions reported to cause intermediary and high risk of mortality when landed using a large (6/0) hook as compared to a small (1/0) hook. Expected Chinook and pink salmon mortality risk scores did not appear to be affected by the size of hook used in recreational troll fishing. The finding that expected Chinook and pink salmon mortality risk was unaffected by hook size was likely driven by low sample sizes in this study, and stronger differences may be elucidated with more robust sample sizes, as seen with coho salmon in this study.

One of the best determinants found for predicting whether a hooked fish will die due to its injury is whether or not the fish is actively bleeding (Cowen, Trouton & Bailey, 2007; Lunzmann-Cooke et al., 2024). We were surprised to find strong evidence that pink salmon hooked on a 1/0 hook were more likely to be observed bleeding than pink salmon hooked on a 6/0 hook. This finding was in contrast to our results for coho and Chinook salmon, in which we found no evidence that hook size affected whether or not a hooked salmon was observed bleeding. Our results of bleeding frequency for pink salmon may have been influenced by our small sample size (14 on 1/0 and 18 on 6/0), but could have also been due to differences in the location of arteries and veins between the salmon species or in their behavior in how they attack a lure. Landed pink salmon were often observed bleeding when hooked in their lower jaw by a small (1/0) hook; bleeding was not commonly observed in other salmon hooked in the lower jaw. More investigation is needed to determine if pink salmon have higher mortality when hooked with a small (1/0) hook than with a large (6/0) hook in recreational salmon troll fisheries given our observation of more frequent observations of bleeding when hooked with the 1/0 hook.

A key finding in this study was that small (1/0) hooks had significantly greater drop-off rates than large hooks. Consideration of drop-off rates is important because hooked salmon that drop off the hook prior to landing can have hooking-related injuries yielding mortality (Lawson & Sampson, 1996). This study likely over-reported the drop-off rate by an unknown amount for both 1/0 and 6/0 hooks because some of the fish that were hooked and lost prior to observation were potentially of a non-salmon species. The greater drop-off rate for 1/0 hooks may have been caused by two factors. The first is the difference in hook strength between the 1/0 and 6/0 hooks. Past research has shown that it takes less force to deform or unbend a 1/0 hook than a 6/0, particularly when the smaller hook has a thinner diameter of wire (Varghese et al., 1997; Edappazham, 2010). We observed multiple instances in which the 1/0 hook we were fishing had bent and opened up, likely increasing the probability of losing a hooked salmon. We were unable to determine when the hooks bent and thus could not evaluate if the bending of the hook had affected the occurrence of drop-offs or how long we fished with the bent hook. We also had a couple instances in which the 1/0 hook broke while reeling up causing the loss of the hooked fish. The second factor that affected drop-off rates is the region of the body that the hook penetrated the fish. A hook that penetrates deeper in the mouth may be more likely to hold the fish and the results of this study show that the 6/0 hook more frequently penetrated anatomical regions deeper in the mouth such as the eye and the cheek than did the 1/0 hook. A follow up study using different size hooks with the same gauge of metal would be beneficial to elucidate what portion of the difference in drop-off rates was due to the shape and size of the smaller hook as compared to the integrity and strength of the metal. Our observed drop-off rates for both 1/0 and 6/0 hooks were lower than the 55.1% drop-off rate observed by Diewert, Nagtegaal & Patterson (2002) during recreational salmon trolling in Canada.

In this study, mean lengths of salmon landed were consistently longer for salmon caught on small (1/0) hooks than salmon caught on large (6/0) hooks for Chinook, coho, and pink salmon (Table 2), but the differences were not statistically significant. This finding was surprising given that the large (6/0) hook had more structural strength (Varghese et al., 1997; Edappazham, 2010) and that past studies have found that larger hooks catch larger fish (Orsi, 1987; Orsi, Wertheimer & Jaenicke, 1993; Wilde, Pope & Durham, 2003; Lindsay et al., 2004; Alós et al., 2008b; Ateşşahin & Dürrani, 2023). It was also surprising to find that hook size had no observable effect on the proportion of catch of legal and sublegal size classes for coho and Chinook salmon (Alós et al., 2008a). Bait shops often tell anglers to buy larger hooks to avoid catching sublegal salmon. Our finding of no difference may have been affected by a low catch rate of sublegal sized salmon during the years of the study resulting in too low of a sample size to detect an effect of hook size on size selectivity for salmon.

Finding potential, easily implemented modifications to fishery regulations to improve the survival of hooked and released salmon is important for salmon conservation (Losee, 2022). Past research has found that a number of factors affect the probability that a hooked fish survives the encounter and that stress is minimized (Muoneke & Childress, 1994; Hühn & Arlinghaus, 2011; Twardek et al., 2018). In this study, we have shown that hook size contributed to the mortality risk of hooked and released coho salmon. However, this study relies on the assumption that mortality risk by body region for salmon caught during a recreational troll fishery using barbless hooks is similar to mortality rates observed during a commercial troll fishery that used barbed 6/0 hooks. We encourage that a follow up study be conducted that directly assesses the mortality rate of hooked and released salmon caught in recreational troll fisheries with post-capture monitoring through tagging or holding and monitoring of the fish. Furthermore, we encourage that a modeling exercise be conducted to determine if the reductions of mortality of hooked and released salmon achieved by using small hooks are offset by the numbers of salmon that incur hooking injuries and are not landed.

Conclusions

This study aimed to determine whether small and large hooks used in recreational troll fishing yield disproportionate risks of mortality to salmon species in the state of Washington. We provide evidence that coho salmon caught on 6/0 hooks are more susceptible to being hooked in body regions reported to have high and intermediary risk of mortality as compared to those caught on 1/0 hooks. Our results and those published by Lunzmann-Cooke et al. (2024), highlight the importance of studying post-release mortality of hooked and released salmon during recreational fisheries to elucidate factors such as hook size that can be regulated to improve post-release survival. The results of this study will hopefully stimulate more research on the effects of hook size in recreational salmon troll fisheries on post-release mortality and evaluations on whether management measures to regulate hook size would benefit salmon conservation.

We would like to thank J Scordino, N Scordino, L Scordino, K Scordino, J Scordino, A Akmajian, C Charles, T Buckingham, J Johnson, T Engram, E Allyn, A Acevedo-Gutierrez, E Acevedo, Z Lewis, A Cunningham, C Richards, R Svec, S Wershow, P DiLella, M Liberatore, D Little, D Little, D Little, B Ayers, X Russell, N Della Smith, L Jimmicum, A Ellis, and L Barnett for volunteering as anglers during this study. We also thank J Roche for making the head region reference graphic. Discussions with T. Burlingame helped with gear arrangement. H Leon provided a review of the manuscript and discussions on the literature.

Additional Information and Declarations

Competing Interests

Author Contributions

Animal Ethics

Data Availability

1 means ’to discover’ or ’seeking knowledge’ in the Makah language.

Jonathan J. Scordino, Ryan P. Walsh, Deon J. Roche, William Jasper are employees of Makah Fisheries Management —Makah Tribe.

Jonathan J. Scordino conceived and designed the experiments, performed the experiments, analyzed the data, prepared figures and/or tables, authored or reviewed drafts of the article, and approved the final draft.

Ryan P. Walsh performed the experiments, analyzed the data, prepared figures and/or tables, authored or reviewed drafts of the article, and approved the final draft.

William Jasper performed the experiments, authored or reviewed drafts of the article, and approved the final draft.

Deon J. Roche performed the experiments, authored or reviewed drafts of the article, and approved the final draft.

William Tyler performed the experiments, analyzed the data, authored or reviewed drafts of the article, and approved the final draft.

The following information was supplied relating to ethical approvals (i.e., approving body and any reference numbers):

This project was reviewed by UC Davis and was found not to require IACUC approval (review number 23171).

The following information was supplied regarding data availability:

The data is available at Mendeley Data: Scordino, Jonathan; Walsh, Ryan; Jasper, William; Roche, Deon; Tyler, William (2024), “Evaluation of hook size in recreational salmon troll fisheries in the marine waters of Washington state, 2023 and 2024”, Mendeley Data, V2, doi: 10.17632/kvdkm7ks6g.2.

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
