# Peer review of "Expected mortality risk for coho salmon landed in recreational troll fisheries using 1/0 and 6/0 hooks in the marine waters of Washington state"

_PeerJ, doi:10.7717/peerj.19434_

## Round 0.1 · original submission · Major Revisions

Dear Dr. Scordino

Expert reviewers have pointed out several critical errors, notably problems with material and methods sections or not citing recent literature.
You can find the comments and suggestions of the expert reviewers in the attached reports. As you will see, some reviewers recommended rejection of the MS, but I think the manuscript has a chance for revision. Consequently, a major revision is needed for your article.

I request you check and correct the manuscript according to the reviewers' reports (please take into account and focus on the comments of reviewer 1, who recommended rejection of your manuscript, in your revision).

Reviewer 1 ·

Basic reporting

In the study, 599 fish were caught over 55 hours of fishing using two different hook sizes. Of these, 187 salmon were captured with 6/0 hooks (35 Chinook, 134 Coho, and 18 Pink Salmon), and 162 salmon with 1/0 hooks (35 Chinook, 113 Coho, and 14 Pink Salmon). The drop-off rate for fish hooked on 1/0 hooks was found to be 1.8 times higher than for those hooked on 6/0 hooks. Significant differences were observed in the anatomical hooking locations of Coho salmon, with 6/0 hooks showing a greater likelihood—approximately double—that fish would be hooked in areas associated with higher mortality risk than 1/0 hooks.

The study observed bleeding, particularly in the head and mouth areas, on salmon caught with two types of hooks, some of which were landed and others released. Different salmon species and quantities were captured using these hooks, drop-off rates were measured, and the potential mortality risk for fish that escaped into the wild was assessed.


The main limitation of the study is that it is unable to assess the mortality risk and survival rate of fish that either escaped the fishing line or were released back into the water from the boat. Another limitation of the study is the exclusion of other commonly used hook sizes.

Experimental design

However, no studies on escape or discard mortality have been conducted, leaving the concept of mortality risk largely unresolved. A more rigorous study design incorporating a control group, tank or net cage observations, and a minimum one-week observation period for three experimental groups per species could have produced more robust findings.

Validity of the findings

Although a significant number of fish were sampled as part of the study, it is not appropriate to base mortality risk assessments on these samples.

Additional comments

Unfortunately, the present study lacks sufficient scientific rigor. Given the importance of this subject, a properly designed, long-term study is essential. In its current form, I believe this study is not suitable for publication.

Reviewer 2 ·

Basic reporting

It is not easy to provide data in recreational angling, but major adjustments are given if adjustments are made within the article.
Sources are very old. articles should be rewritten with some current articles.

Experimental design

The section on material and method in the text needs to be corrected.

Validity of the findings

The findings are well prepared.

Annotated reviews are not available for download in order to protect the identity of reviewers who chose to remain anonymous.

Reviewer 3 ·

Basic reporting

A)"The translation of the text has generally been done accurately and meaningfully. However, I have a few minor grammatical and stylistic corrections to suggest, which may enhance its academic tone and clarity. By taking these suggestions into account, we can make the text more fluid and precise. Here are some points to consider:"
1.In scientific contexts, "size" is often replaced with "size class" or "size distribution" when referring to the physical dimensions of fish. So, "size of salmon caught" can be better translated as "size class of the salmon caught" or "size distribution of the salmon caught".
2.The phrase "risk of mortality" is a standard term in scientific studies. The expression is grammatically correct, but sounds more natural in this context.
3.This sentence can be clarified in English as:
"Although no significant differences were observed in the hooking rates of Chinook and pink salmon in body regions with high, intermediate, or low mortality risk, the sample size limits our confidence in this finding."
This revision clarifies that it refers to body regions and eliminates ambiguity.
4. A more fluent translation in English could be:
"It is important to carefully assess whether these results support restrictions on hook sizes used in recreational coho salmon troll fisheries."
B) Improving the Clarity and Flow of the Writing
The overall language is generally academic and correct, but a few sections could benefit from a more fluid and clearer expression. Here are a few suggestions:
• Introduction to the Study:
Instead of saying, “This study evaluates the fishing performance of small and large hooks in recreational trolling based on three parameters,” you could say:
"This study investigates the fishing performance of small (1/0) and large (6/0) hooks in recreational trolling fisheries, focusing on three key parameters."
• Clarifying the Purpose of the Study:
In addition to stating the specific aims, it could help to give a clearer statement of what the study hopes to achieve overall:
"The overall objective of this study is to evaluate how hook size influences the post-release mortality risk, drop-off rates, and the size of salmon caught in recreational coho and chinook fisheries."
• Linking Hypotheses to Management Implications:
After stating the hypotheses, briefly explaining their potential implications for fishery management can add weight to your research. For instance:
"If hook size is found to reduce mortality and drop-off rates, it could lead to new management recommendations regarding hook regulations in recreational salmon fisheries."
C) Strengthening Connections to Previous Research and Literature
• When presenting your hypotheses, it’s helpful to tie them more explicitly to previous studies. For example, referencing how your hypothesis aligns or contrasts with the findings of Vincent-Lang, Alexandersdottir & McBride (1993) or other studies would strengthen the scientific foundation for your predictions.
“While studies such as Vincent-Lang et al. (1993) suggest that smaller hooks may reduce post-release mortality, our hypothesis extends this by specifically testing the effects on anatomical regions with varying mortality risks.”
• Clarifying the relationship between your study and existing literature can also highlight the novelty and importance of your work. Consider adding a sentence like:
“This study builds upon previous research that has shown varying results regarding hook size, providing further insight into the potential benefits for fishery management.”
D) Strengthening Connections to Previous Research and Literature
• When presenting your hypotheses, it’s helpful to tie them more explicitly to previous studies. For example, referencing how your hypothesis aligns or contrasts with the findings of Vincent-Lang, Alexandersdottir & McBride (1993) or other studies would strengthen the scientific foundation for your predictions.
“While studies such as Vincent-Lang et al. (1993) suggest that smaller hooks may reduce post-release mortality, our hypothesis extends this by specifically testing the effects on anatomical regions with varying mortality risks.”
• Clarifying the relationship between your study and existing literature can also highlight the novelty and importance of your work. Consider adding a sentence like:
“This study builds upon previous research that has shown varying results regarding hook size, providing further insight into the potential benefits for fishery management.”
E) Framing the Results and Interpretation
• Framing the results: When presenting the results, it’s important to go beyond just stating statistical significance. You could briefly interpret what the results imply for management practices or the broader ecological context. For example:
"If our results confirm that larger hooks reduce post-release mortality in coho and chinook salmon, it could support regulations favoring larger hooks in recreational fisheries to improve salmon survival."
F) Suggestions for Future Research
• Including a “Future Research” section in your conclusion would be beneficial. Suggesting specific areas where more research is needed would help other researchers build on your work. For instance:
"Future research should focus on direct comparisons of multiple hook sizes and explore the long-term impacts of reduced hooking mortality on salmon population recovery, using tagging and monitoring techniques."

Experimental design

1. Refining and Strengthening the Hypotheses
The hypotheses are generally valid, but they could be more detailed and framed more robustly in scientific terms. This will improve their testability and make the findings more conclusive. Below are suggestions for refining each hypothesis:
First Hypothesis:
Current: "We hypothesized that small hooks would have significantly lower frequency of hooking body regions with intermediary or high risk of post-release mortality."
Suggested Revision: "We hypothesized that small hooks would hook salmon in body regions with medium or high post-release mortality risk less frequently than large hooks."
Why the change?
• This revision clarifies the hypothesis and emphasizes the specific anatomical regions with higher post-release mortality risk. It makes the statement more straightforward and scientifically precise.
Second Hypothesis:
Current: "We evaluated if using a small hook would increase drop-off rates compared to large hooks."
Suggested Revision: "We hypothesized that small hooks would result in higher drop-off rates compared to large hooks."
Why the change?
• Instead of phrasing it as an evaluation, stating it as a clear hypothesis ("would result in") makes the intention more direct and testable. It sets up a clear expectation that can be evaluated during the study.
Third Hypothesis:
Current: "We hypothesized that we would catch larger salmon on larger hooks and reduce hooking rates on sub-legal sized coho and chinook salmon."
Suggested Revision: "We hypothesized that larger hooks would catch larger coho and chinook salmon and reduce the hooking rates of undersized (sub-legal) coho and chinook salmon."
Why the change?
• Rewording "catch larger salmon on larger hooks" as "larger hooks would catch larger coho and chinook salmon" is more direct. Also, specifying "undersized (sub-legal)" makes it clear that these are fish below legal limits for retention, reducing any potential ambiguity.
Strengthening the Experimental Methods and Data Analysis
• Clarifying the experimental design and statistical analysis methods directly linked to the hypotheses would further strengthen the study. For example, explicitly stating the statistical tests or analytical models you plan to use to test the hypotheses can add more confidence to the research framework.
• Clarifying the hook size definitions: The terms “small” and “large” hooks are somewhat general. It would help if you clearly define the specific hook sizes used in the study (e.g., 1/0 and 6/0) when discussing the experimental design. This makes the study more reproducible and ensures transparency.

Validity of the findings

Suggestions for Improvement:

Clarifications on Measurements and Data Collection:

A more detailed explanation of how the fork length (FL) measurement is taken (e.g., from the tip of the head to the tail notch?) could be provided. This would help ensure the consistency of the measurements.
Details on Fish Handling Methods:

Adding more information on how the fish were handled (e.g., how quickly they were released after being measured) would enhance the reliability of the methodology, particularly considering that post-handling survival rates may be affected.
Potential Bias in Double Hooking Situations:

It may be helpful to provide further details on the anatomical regions where the fish were hooked. If a fish was hooked in multiple areas, including information on the protocol followed in such cases would improve the credibility of the research.
Environmental and Temporal Variables:

Providing additional information on environmental conditions (e.g., water temperature, salinity, or weather) would be valuable for understanding how these factors might influence the fish's reaction to the hook or the overall fishing success.
Detection Method for Hooking:

The explanation of the "hooking detection" method could be more detailed. Additional information on how the movement at the tip of the rod was observed, as well as whether false positives or negatives were possible with this method, would be beneficial.

Additional comments

"By reviewing the points outlined below, the quality of the paper will be improved.

1. Strengthen the Summary and Introduction
• Clarify the purpose and significance of the study: A lot of the provided information is methodological, but for a reader to quickly grasp the main goal and importance of the study, you could include a brief introduction that highlights this. Especially answering the question "Why was this study done?" will help make the context and significance clearer for readers.
• Suggested introductory sentence:
"The primary objective of this study was to compare the fishing performance of small (1/0) and large (6/0) hooks in recreational salmon troll fisheries in Washington state. Understanding the effects of hook size on salmon size, mortality risk, and fishing efficiency is crucial for sustainable fisheries management."

2. Make the Methodology Section More Organized and Fluid
• Consolidate excessive technical details and organize them: The methodology section is quite technical, so it could be made smoother and more readable by linking certain parts more logically. For example, transitions between equipment used and procedures could be softened, and you could use more connecting phrases to help the reader follow the flow.
• Suggested improvement:
"In this study, we compared the performance of two hook sizes (1/0 and 6/0) using the same brand of hook (Gamakatsu Octopus) for both sizes. This approach ensured that any observed differences in fishing performance were due solely to hook size, and not to variations in design between different manufacturers."

3. Simplify and Clarify the Data Analysis Section
• Add more explanations: You could make your statistical analyses clearer by adding brief explanations of why certain tests were used. For example, explaining why you chose Chi-squared tests or how you decided on specific test parameters will help the reader understand your approach better.
• Suggested improvement:
"Chi-squared tests were used for data analysis as this test is appropriate for comparing categorical variables. It allowed us to examine whether the proportion of fish caught by hook size differed. Drop-off rates were similarly compared using a Chi-squared test to determine if hook size had any significant effect on fish escape rates."

4. Clarify the Assumptions and Limitations of the Study
• Highlight assumptions and limitations: Like any study, there are assumptions and limitations. Especially with your mortality risk estimates, it is important for readers to understand the assumptions you are making. Making these clear will help contextualize the results.

5. Emphasize Results and Findings Earlier
• Introduce key findings earlier: Introducing your results early in the text, even in the methodology section, can help to clarify the context and significance of your study. This will make the reader better understand the goals and the relevance of the methods you are using.
• Suggested improvement:
"The results indicated that smaller hooks (1/0) may reduce mortality rates but lead to higher drop-off rates. These findings suggest that while smaller hooks might enhance salmon survival, they may also increase the number of fish that escape during retrieval."

6. Strengthen the Academic Tone
• Use more academic language: Some sentences can be rewritten in a more formal, academic tone. This will make the paper sound more scientific and improve its credibility.

Best regards, and I look forward to your continued efforts."

---

## Round 0.2 · accepted · Accept

Dear Dr. Scordino,

I thank you for making the corrections and changes requested by the reviewers. I read and checked your valuable article carefully and am happy to inform you that the article has been accepted for publication in PeerJ.
Sincerely yours,

Reviewer 2 ·

Basic reporting

Some of the necessary arrangements were made in line with the criticisms made. However, some edits were not made. According to the opinions of other referees and my own opinions, this article has reached a publishable stage. The addition of new sources will be important for the article to be more readable. Best regards.

Experimental design

Although there are shortcomings, it is generally organized.

Validity of the findings

It looks better after the corrections.

Reviewer 3 ·

Basic reporting

Thank you for your detailed and thoughtful response to the reviewer comments. It is evident that significant effort was made to revise the manuscript both in terms of content and structure. Your clarifications regarding the use of previously published mortality risk data, your discussion on resource limitations, and your rationale for the chosen hook sizes have made the study’s scope and contributions clearer.

Experimental design

The improved framing of hypotheses, enhanced methodological transparency, and inclusion of more recent references have strengthened the manuscript considerably.

Validity of the findings

While limitations related to direct mortality observation still exist, your acknowledgment of these issues and suggestion for future research are scientifically appropriate and transparent.

Additional comments

Overall, I believe your revised manuscript is now suitable for publication. Congratulations on your work.